# Biodegradable Nanofiber/Metal–Organic Framework/Cotton Air Filtration Membranes Enabling Simultaneous Removal of Toxic Gases and Particulate Matter

**DOI:** 10.3390/polym15193965

**Published:** 2023-09-30

**Authors:** Sujin Ryu, Doyeon Kim, Hyewon Lee, Yoonjin Kim, Youngbok Lee, Myungwoong Kim, Heedong Lee, Hoik Lee

**Affiliations:** 1Advanced Textile R&D Department, Research Institute of Convergence Technology, Korea Institute of Industrial Technology (KITECH), 143 Hanggaulro, Sangnok-gu, Ansan 15588, Republic of Korea; sjryu@kitech.re.kr (S.R.); dobbie@kitech.re.kr (D.K.);; 2HYU-KITECH Joint Department, Hanyang University, Ansan 15588, Republic of Korea; yblee@hanyang.ac.kr; 3Department of Applied Chemistry, Hanyang University, Ansan 15588, Republic of Korea; 4Department of Chemical and Molecular Engineering, Center for Bionano Intelligence Education and Research, Hanyang University, Ansan 15588, Republic of Korea; 5Department of Chemistry and Chemical Engineering, Inha University, Incheon 22212, Republic of Korea

**Keywords:** nanofiber, cellulose, biodegradable, metal organic frameworks, filtration

## Abstract

The typical filters that protect us from harmful components, such as toxic gases and particulate matter (PM), are made from petroleum-based materials, which need to be replaced with other environmentally friendly materials. Herein, we demonstrate a route to fabricate biodegradable and dual-functional filtration membranes that effectively remove PM and toxic gases. The membrane was integrated using two layers: (i) cellulose-based nanofibers for PM filtration and (ii) metal–organic framework (MOF)-coated cotton fabric for removal of toxic gases. Zeolitic imidazolate framework (ZIF-8) was grown from the surface of the cotton fabric by the treatment of cotton fabric with an organic precursor solution and subsequent immersion in an inorganic precursor solution. Cellulose acetate nanofibers (NFs) were deposited on the MOF-coated cotton fabric via electrospinning. At the optimal thickness of the NF layer, the quality factor of 18.8 × 10^−2^ Pa^−1^ was achieved with a filtration efficiency of 93.1%, air permeability of 19.0 cm^3^/cm^2^/s, and pressure drop of 14.2 Pa. The membrane exhibits outstanding gas adsorption efficiencies (>99%) for H_2_S, formaldehyde, and NH_3_. The resulting membrane was highly biodegradable, with a weight loss of 62.5% after 45 days under standard test conditions. The proposed strategy should provide highly sustainable material platforms for practical multifunctional membranes in personal protective equipment.

## 1. Introduction

Numerous air pollutants, such as hazardous chemicals and particles, have been released from industrial facilities and other human activities and may adversely affect the environment and human health [1]. The World Health Organization (WHO) has reported that approximately seven million people have died due to exposure to air pollution, suggesting that air pollution is a critical health risk [2]. In particular, particulate matter (PM) is currently considered a significant air pollutant that can potentially lead to respiratory disease [3,4]. Other hazardous contaminants are toxic gases that can eventually displace oxygen from the lungs or can damage the respiratory system, e.g., volatile organic compounds (VOCs), aldehydes, H_2_S, NH_3_, and so on [2,5,6]. In addition, exposure to air, including sources for infectious and major communicable diseases, such as COVID-19 is highly problematic. Therefore, protection from the above-mentioned pollutants requires the use of personal protective equipment (PPE), such as face masks that have become indispensable.

Membrane filters to block air pollutants are necessary for filter-type PPE, which is generally made from synthetic polymers, such as polypropylene (PP) and polyethylene terephthalate (PET). Although these synthetic polymers are highly processable and economical, their high stability has resulted in other environmental problems. For example, the mandatory use of face masks during the COVID-19 crisis has posed a strong challenge to plastic waste management and subsequent adverse environmental effects [7]. According to the WHO, the weight of PPE purchased worldwide due to COVID-19 in 1.5 years was approximately 87,000 tons, eventually ending in landfill [8]. Along with the dramatic increase in the demand for PPE, eco-friendly membrane filters with materials that can replace synthetic polymers are needed.

Cellulose is an abundant and biodegradable natural polymer material exhibiting outstanding mechanical, chemical, and biological properties and is advantageous in various applications, such as composite science [9], paper engineering [10,11], and medical engineering [12]. Cotton is a well-known cellulose-based fabric material utilized in clothing. Despite the advantages of cotton, it has not been used as a component for filtration devices because of its large pore size and incompatibility with the corona charging process, which is a typical method applied widely to PP non-woven filters for removing PM. To apply a conventional cotton fabric as a filtration media, nanofabrication can be an effective approach as it has been a workhorse in the field of air filtration [13,14]. In particular, electrospinning is a representative technique to produce ultra-fine polymeric nanofiber (NF) mat in a scale of from several tens to hundreds of nanometers [15], resulting in high surface area and a sub-micron sized pore structure, which is highly advantageous in various applications including filtration and biomedical engineering [16,17]. The effective combination, i.e., stacking NF on the cotton fabric, is feasible for improving the filtration performance against PM.

In addition, the chemical structure of the cellulose is beneficial to tailor its properties to achieve the target application because it has abundant multiple hydroxyl groups providing useful reactivity to incorporate various chemical functions for desirable properties [18,19]. The metal–organic framework (MOF) is a widely used material as an adsorbent for different gas molecules due to its high specific surface area and porosity [20]. Among the various MOF materials, the zeolitic imidazolate framework (ZIF) has a high potential for the removal or storage of specific gas molecules [21,22]. The incorporation of ZIF onto a cotton fabric possibly provides an effective route for MOF-incorporated cotton fabric that is applicable for removing toxic gases. Hydroxy groups in the cotton fabric can effectively interact with precursor molecules to synthesize ZIFs, allowing direct growth on the surface of the cotton fabric, eventually leading to a filtration membrane capable of adsorbing toxic gases.

The multifunctional fibrous membrane for removing particulate matter and toxic gases was demonstrated by employing simultaneous electrospinning/electrospray of poly(vinyl alcohol) and MOF onto PET nonwoven fabric [4]. However, it does not only require a complex co-electrospinning system, but also in the process, the adsorbed MOF particles can be desorbed in a variety of conditions. More importantly, the system is yet biodegradable, which is not preferred as a next-generation material platform. Herein, we report the fabrication route of a biodegradable cellulose-based filtration membrane via the direct growth of ZIF-8 from the surface of cotton fabric and subsequent electrospinning of cellulose acetate (CA) solution to form a CA NF/MOF-coated cotton fabric membrane that can filter PM and toxic gases simultaneously (Figure 1). The major components of the membrane are made from cellulose-based materials to ensure biodegradability. ZIF-8 was selected as a coating material on the cotton fabric to remove toxic gases, e.g., H_2_S, formaldehyde, and NH_3_ [22]. The filtration performance of the resulting membrane was assessed by controlling the amount of NF deposited on the MOF-coated cotton fabric, maximizing the quality factor towards the optimal filtration performance parameters. The multi-layered membrane effectively removed the three model toxic gas molecules. In addition, the membrane showed adequate biodegradability, highlighting its potential to replace the current petroleum-based polymer membrane filters.

## 2. Materials and Methods

### 2.1. Materials

Cellulose acetate (CA; M_n_ ≈ 50,000 g/mol), cellulose (particle size ≈ 20 μm, as a standard material of biodegradability), *N*,*N*-dimethylformamide (DMF; 99.8%), acetone (≥99.5%), zinc nitrate hexahydrate (98%), and 2-methylimidazole (99%) were purchased from Sigma-Aldrich Co., Ltd. (St. Louis, MO, USA). The cotton fabric (30 g/m^2^) was purchased from Kolon Industries Inc. and was cut to a dimension of 20 cm × 28.5 cm for use in the fabrication of a filtration membrane. All chemical reagents were used as received.

### 2.2. Microscopic Characterization

The morphologies of the NFs on cotton fabrics were examined using field-emission electron microscopy (FE-SEM; SU8010, HITACHI, Tokyo, Japan).

### 2.3. Spectroscopic Characterization

X-ray diffraction (XRD) studies were carried out using a RINT2000 X-ray diffractometer (Rigaku, Tokyo, Japan) using a Cu Kα source. Fourier-transform infrared (FT-IR) spectra were recorded on Perkin Elmer Frontier in the wavenumber range of 400–4000cm^−1^ (Waltham, MA, USA).

### 2.4. Assessment of Air Permeability

Air permeability was estimated using FX 3300 Air Permeability Tester III (TEXTEST, Schwerzenbach, Switzerland) by measuring an air flow rate perpendicularly passing through the sample (effective area = 38 cm^2^) under an air pressure difference of 125 Pa [23]. The particulate respiration filter test was performed using a TSI Automated Filter Tester 8130A (TSI, Shoreview, MN, USA) to examine the filtration efficiency and pressure drop using a sodium chloride challenge aerosol at a flow rate, i.e., a face velocity of 32 L/min. When the aerosol passed through the filter, the aerosol was measured simultaneously using a light scattering photometer. Five measurements were performed, and the average and the standard deviation values were reported. A filtration efficiency (η) that represents the filtration performance of the membrane was estimated using Equation (1), and a quality factor (Q_f_) was obtained using Equation (2):η = (C_up_ − C_down_)/C_up_(1)
where concentrations are upstream (C_up_) and downstream (C_down_).
Q_f_ = −ln(1 − η)/∆P(2)
where ∆P is the pressure drop across the filtration medium [24].

### 2.5. Evaluation of Gas Adsorption Capability

The adsorption capability for toxic gases was measured using a gas detector tube method. The membrane sample (10 cm × 10 cm) was placed in a chamber (2 L) filled with a gas of interest at 100 ppm. The gas concentration was measured using a valve for up to an hour. The rate of the gas concentration decrease (CDR) was determined using Equation (3):CDR (%) = [(C_i_ − C_f_)/C_i_] × 100(3)
where C_i_ and C_f_ are concentrations at the initial and final states, respectively.

### 2.6. Assessment of Biodegradability

The biodegradability was assessed using the procedure and conditions following KS M ISO 14855-1. Standard compost (ABNEXO, Gyeonggi, Republic of Korea; dry solid content = 52.5%) was used as an inoculum. Cellulose was used as a standard material, and NF/MOF-coated cotton samples were used as test materials after cryo-milling to less than 250 μm. The mixture of standard compost with a moisture content of 50% (300 g), standard material (25 g), or test material (25 g) was placed in a composting container, followed by the measurement of CO_2_ generation every 24 h for 45 days. Biodegradability (D_t_) was calculated using Equation (4):D_t_ = [((CO_2_)_T_ − (CO_2_)_B_)/(CO_2_)_Th_] × 100(4)
where the (CO_2_)_T_ and (CO_2_)_B_ are the cumulative amount of CO_2_ generated from the composting container containing the test material and standard material, respectively, and (CO_2_)_Th_ is the theoretical amount of CO_2_ generated from the test material.

### 2.7. Preparation of MOF-Coated Cotton Fabric

An aqueous solution of 2-methylimidazole (0.24 M) was prepared for use as a ligand solution. Cotton fabric was immersed in a ligand solution for 3 h at room temperature. The cotton fabric was applied to the padder (DL-2005, Daelim starlet, Siheung, Republic of Korea) at 0.5 bar and 1 m/min. The resulting cotton fabric was immersed in an aqueous solution of zinc nitrate hexahydrate (0.48 M) for 5 h, and dried under ambient conditions at room temperature for 24 h.

### 2.8. Electrospinning of CA NFs on Cotton Fabric

CA NFs were fabricated on the prepared MOF/cotton fabric using an electrospinning process. The electrospinning apparatus was equipped with a high-voltage power supply (NC-ESR100S, NanoNC, Seoul, Republic of Korea) as a source of the electric field, a plastic syringe with a metallic needle (25 gauge), and a cylinder-type metallic drum as a collector. A 19 wt% CA solution in a mixed solvent of DMF and acetone (4/6, *v*/*v*) was prepared via vigorous stirring at room temperature for 24 h. The solution was injected into the syringe and subjected to electrospinning at a voltage of 15 kV and a flow rate of 10 μL/min with a tip-to-collector distance of 15 cm. The cotton fabric was mounted on the metallic drum collector using scotch tape. The funnel collector rotated at the speed of 100 rpm. All electrospinning experiments were conducted at room temperature with a relative humidity of 35–40%.

## 3. Results and Discussion

### 3.1. Fabrication of MOF-Coated Cotton Fabric

A multifunctional biodegradable membrane filter was fabricated by stacking CA NFs onto the MOF-coated cotton fabric. The cotton fabric was used as a supporting material due to its outstanding mechanical properties, biodegradability, and inexpensiveness. More importantly, the abundant hydroxyl groups in the cellulose enable the formation of MOF structure on the surface of the cotton. The adsorption of the ligand readily occurs when the cotton fabric is immersed in the ligand solution because the hydroxyl group can effectively interact with 2-methylimidazole [25]. After immersion, the padding process was carried out to remove the excess ligands not adsorbed on the fabric surface. The resulting cotton fabric was finally immersed in a metal ion precursor solution to form ZIF-8 MOF particles. First, the effect of the variation in the amounts of reactants, i.e., Zn^2+^ and ligand, on the morphology of resulting MOF particles was assessed without cotton (Appendix A). Upon varying the [ligand] to [Zn^2+^] ratio from 2.0 to 8.0, the morphology of the MOFs fabricated with a ratio of 2.0 led to 1–2 μm spherical particles. The morphology was changed to a disc-like or elliptical structure as the ratio increased [26]. In addition, at a ratio higher than 4, the size of disc-like particles gradually decreased, which is in good agreement with the literature [27]. For the applicability to remove gas species, [ligand]/[Zn^2+^] was set to 2.0 to attain a small spherical MOF structure and maximize the adsorption capability and surface area. ZIF-8/cotton was fabricated at a [ligand]/[Zn^2+^] ratio of 2.0, and the morphologies were examined (Figure 2). The surface of the pristine cotton fabric was very smooth (Figure 2a). In contrast, the MOF/cotton fabric exhibited some grains on its surface, indicating the formation of the MOF particles on the surface (Figure 2b). In addition, the resulting MOF/cotton fabric was washed sufficiently with ethanol, acetone, and water to remove weakly attached particles, and strongly attached MOF particles were observed on the surface (Figure 2c), which is of importance to ensure the stability of the ZIF-8 coating on the fabric.

### 3.2. Electrospinning of CA Nanofiber on MOF-Coated Cotton Fabric

On the MOF-coated cotton membrane, a CA solution was electrospun at fixed concentration with different amounts of solution from 0.6 mL to 7.2 mL. Although the concentration affects the fiber diameter in the electrospinning process [28,29], in terms of filtration efficiency, the thickness of the nanofiber mat is more crucial [4,30]. Thus, we focused on the effect of the thickness of the nanofiber mat fabricated at the fixed concentration of the polymer solution. Figure 3 presents the morphologies of CA NFs on MOF/cotton fabrics, namely NF_0.6_, NF_1.8_, NF_3.6_, NF_6.0_, and NF_7.2_, where the number indicates the volume (mL) of electrospun solution. All membranes exhibited a straight nanofibrous structure with high uniformity and smooth surface on cotton fabrics, suggesting that the amount of electrospun solution did not affect the morphology, as expected. The thicknesses of the resulting NF membranes were 0.222, 0.232, 0.253, 0.279, and 0.293 mm for NF_0.6_, NF_1.8_, NF_3.6_, NF_6.0_, and NF_7.2_, respectively. In NF_0.6_, the underlying cotton fabric was observed readily even after NF was formed on its top (Figure 3a), suggesting that the NFs did not sufficiently cover the cotton fabric. In contrast, the NF samples fabricated with the solution with a volume higher than 1.8 mL fully covered the fabric, which was vital for the PM removal filter. The cross-sectional SEM image of NF_7.2_ (Figure 3f) revealed a bilayer structure, confirming that NFs were deposited well on the cotton fabric without a gap between the two layers.

Commercial filters made from synthetic polymers, such as polyethylene (PE) and PP, can be used for filtration applications after treatment with corona charging, one of the most exploited techniques because of its convenience. Under humid or high-temperature conditions, however, the charges injected into the dielectric materials can be dissipated easily, resulting in a degradation of the durability of the resulting membrane [31]. Since cotton can effectively interact with water molecules, despite exhibiting excellent biodegradability, it cannot be used widely as a filter medium because of its high water content. In this perspective, electrospun CA NFs are beneficial as they can impart filtration capability and circumvent this issue. Nano-sized fibers should compensate for the disadvantages of cotton fabric by improving the filtration efficiency much higher than cotton fabrics because of its extremely small pores in the NF membrane. It should be noted that high filtration efficiency can also result in low air permeability and high-pressure drop, which are essential factors in filtration performance. Therefore, optimizing the filtration efficiency and air permeability is essential, which is feasible with the control of the thickness of the nanofiber mat, which is one of the most significant parameters to affect the performance of air filtration membranes.

### 3.3. Characterization of the Fabricated Filtration Membrane

The fabricated composite fibrous membranes were characterized by FT-IR spectroscopy and XRD (Figure 4). In the FT-IR spectrum of MOF (Figure 4a), the peak at ≈760 cm^−1^ was assigned to the stretching vibration mode of Zn-O in ZIF-8 [21]. The peaks at ≈1150 and ≈1310 cm^−1^ were assigned to the C–N bending mode of the imidazole group [21,32,33]. These characteristic peaks were barely observed in the spectra of MOF/cotton and NF/MOF/cotton membranes, which was expected, as the relative amount of MOF in the composite system is very small compared to cotton and NF; compared to cotton fabric, approximately 2.8 wt% of MOF particles were formed, resulting in a very low intensity of MOF in the FT-IR spectra. The peaks due to CA were readily observed in the composite fabric sample. The peaks at ≈1750 cm^−1^ and ≈1220 cm^−1^ were assigned to C=O and C–O bonds of a carbonyl group, which were not found in cotton and MOFs/cotton and only appeared in the CA NF/MOFs/cotton sample [18,19]. Given the configuration of the composite sample, the appearance of the characteristic peaks due to CA NF is acceptable. These results suggest that the CA NF had been deposited effectively on MOF-coated cotton fabric. The resulting sample was characterized by XRD (Figure 4b). In the pattern of only MOF (green trace), the XRD peaks of MOF were observed at 8.0°, 12.0°, 15.0°, and 17.6° 2θ [34,35]. In particular, the peak at 8.0° 2θ (011) was a sharp characteristic crystalline peak typically representing the sodalite structure of ZIF-8 with a body-centered cubic packing. The cotton sample did not exhibit an XRD peak at around 8.0°; some broad peaks at 14.75°, 16.48°, and 22.71° 2θ were observed, indicating the semicrystallinity of cotton fabric [36]. On the other hand, MOF/cotton and CA NF/MOF/cotton membranes exhibited an XRD peak at 8.0° 2θ, suggesting that ZIF-8 had grown on the cotton fabric, and the structure was fully preserved during the electrospinning process.

### 3.4. Air Filtration Performance of the Filtration Membrane

To optimize the filtration performance, we assessed the air permeability, filtration efficiency, pressure drop, and Q_f_ for cotton, MOF-coated cotton, and CA NF/MOF-coated cotton (Figure 5). In addition, the amount of the CA solution that was electrospun onto MOF-coated cotton fabric was varied because it determines the density of CA NF, which is directly related to the porosity of the NF membrane and, hence, the final filtration performance. In the case of MOF-coated cotton and NF_0.6_/MOF-coated cotton membranes, air permeability and pressure drop did not change significantly compared to the pristine cotton fabric. The filtration efficiencies were 19.6, 21.3, and 54.6% for the cotton, MOF-coated cotton, and NF_0.6_/MOF-coated cotton membrane, respectively (Figure 5a). These values are insufficient to filter PM effectively. These results suggest that the growth of ZIF-8 on the surface of cotton and the deposition of small amounts of NF did not result in high enough porosity for practical filtration performance. However, the NF_7.2_/MOF-coated fabric showed a largely increased filtration efficiency of ≈99%, whereas the air permeability was reduced significantly from ≈23 to ≈10 cm^3^/cm^2^/s. In addition to these changes, a pressure drop increased gradually from ≈14 (NF1.8/MOF-coated cotton fabric) to ≈67 Pa (NF_7.2_/MOF-coated fabric). Finding the optimal parameter, i.e., the amount of NF, is critical to attaining the membrane utilizable for PPE because these filtration performance parameters are in the trade-off relation. A quantitative evaluation of the filtration performance can be carried out by estimating Q_f_, which is affected significantly by the thickness of the NF layer that can be tuned readily by controlling the amount of electrospun CA solution. The NF_1.8_/MOF-coated cotton fabric membrane showed the highest Q_f_ value (18.8 × 10^−2^ Pa^−1^) with an air permeability of ≈19 cm^3^/cm^2^/s and pressure drop of ≈14 Pa, which are acceptable for the filtration medium in facemasks [37]. Recently, a dual-functional hybrid filtration system consisting of MOF and NF was reported for the removal of particulate matter and toxic gases in the air [4]. The study highlighted the importance of finding the most desirable NF, MOF, and non-woven fabric configurations for optimal filtration performance for the practical PPE application. The membrane in the current study also shows comparable air filtration performance, ensuring the effectiveness of the proposed fabrication approach. Therefore, the NF_1.8_/MOF-coated cotton membrane exhibiting the highest Q_f_ was further assessed for its ability to remove toxic gases.

### 3.5. Gas Adsorption Behavior of Membrane Filter

Three model gases, i.e., H_2_S, formaldehyde, and NH_3_, known as harmful gases representing acidic, neutral, and basic gases, were tested to assess the capability to remove toxic gases. Figure 6 shows the gas-capturing behaviors of the pristine cotton, MOF-coated cotton, and NF_1.8_/MOF-coated cotton membranes. The CDR values after 60 min for pure cotton fabric showed ≈10%, ≈15%, and ≈30% for H_2_S, formaldehyde, and NH_3_, respectively. On the other hand, MOF-coated cotton fabric showed dramatically improved adsorption capability compared to pristine cotton fabric. MOF is highly effective in capturing these gases because of its active sites and large surface area [38]. In particular, ZIF-8, which was grown as an MOF on the cotton fabric, is used widely for adsorbing harmful gas species [39]. As expected, MOF-coated cotton and NF_1.8_/MOF-coated cotton membranes exhibited a CDR of approximately 99% for all three gases after 60 min. These results strongly suggest that the direct growth of MOF-coated cotton fabric is highly effective for removing toxic gases, even with the NF layer imparting PM filtration capability.

### 3.6. Biodegradability of the Filtration Membrane

The majority of the NF_1.8_/MOF-coated cotton membrane is a cellulose-based material because the calculated weight ratio of the components is 16.2:2.8:81.0 (NF, MOF, and cotton), suggesting that the hybrid membrane exhibits adequate biodegradability. The NF_1.8_/MOF-coated cotton fabric showed comparable biodegradation behavior (Figure 7). The biodegradability of the membrane reached 56.9% after 30 days, converging to 62.5% after 45 days. According to the ASTM standard, a material that can be degraded by more than 60% (weight loss) within 180 months is biodegradable [40]. The cotton-based membrane filter is expected to exhibit biodegradability because cotton is readily degraded by microorganisms in the soil when landfilled. Venditti et al. reported that approximately 40% of cotton fabric was degraded after 40 days [41]. Li et al. reported that the degradation efficiency of the cotton fabric varied in the range of 50–77% after 90 days of treatment [42]. Our results agree well with the literature; therefore, the proposed route effectively fabricates a cellulose-based biodegradable dual-functional membrane to remove PM and toxic gases simultaneously.

## 4. Conclusions

A biodegradable dual-functional hybrid membrane was demonstrated via the route by combining the direct growth of MOF on cotton fabric and the subsequent electrospinning of a CA solution. The ZIF-8 was grown from the surface of a cotton fabric by treatment with an organic precursor solution and subsequent immersion in a metal ion solution. The CA NF was electrospun onto the MOF-coated cotton fabric with different amounts of CA solution to control the porosity of the membrane. The thickness of the NF layer largely affected the filtration performance of particulate matter, as indicated by the assessment of the filtration efficiency, pressure drop, and air permeability. In general, with a thicker NF layer, the filtration efficiency and the pressure drop tended to increase, even though the air permeability decreased. As all of these parameters are in the trade-off relationship: Q_f_ was assessed to find the optimal thickness of the NF layer; NF_1.8_/MOF-coated cotton fabric exhibited the highest Q_f_ value (18.8 × 10^−2^ Pa^−1^) with an air permeability of 19.0 cm^3^/cm^2^/s, a pressure drop of 14.2 Pa, and filtration efficiency of 93.1%. The MOF-coated cotton and NF_1.8_/MOF-coated cotton membrane showed CDR values higher than 99% for H_2_S, formaldehyde, and NH_3_, confirming the achievement of the dual functions of toxic gas removal and the filtration of PM in the membrane system. Finally, the NF_1.8_/MOF-coated cotton fabric membrane showed reasonable biodegradability with a weight loss of 62.5% after 45 days, which is acceptable to consider the system as a biodegradable material. The presented strategy combining the growth of MOF and electrospinning cellulose-based NF should offer an eco-friendly approach to achieve lightweight PPE filters that remove both PM and toxic gases simultaneously in a sustainable manner.

## Figures and Tables

**Figure 1 polymers-15-03965-f001:**
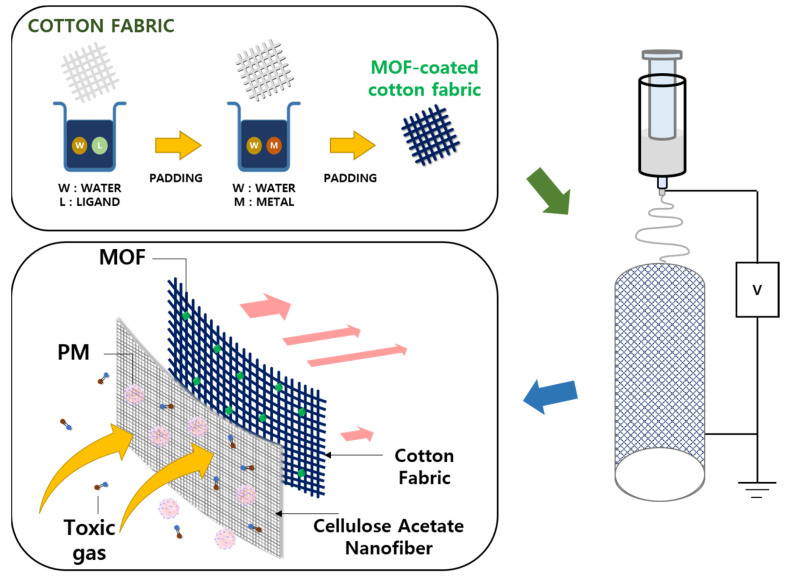
Schematic diagram displaying the fabrication of the multifunctional hybrid CA NF/MOF-coated cotton membrane for the simultaneous filtration of toxic gas molecules and PM.

**Figure 2 polymers-15-03965-f002:**
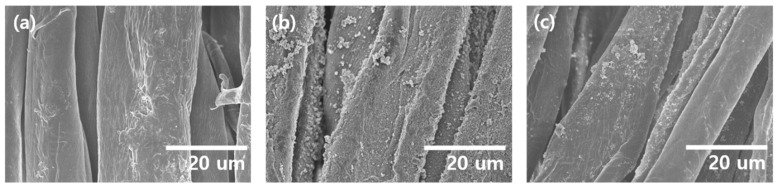
SEM images showing the morphology of (**a**) padded cotton and (**b**) ZIF-8-coated cotton before washing, and (**c**) ZIF-8-coated cotton after washing.

**Figure 3 polymers-15-03965-f003:**
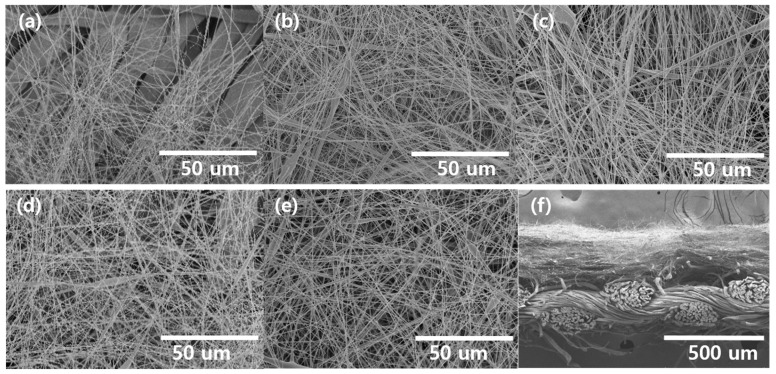
SEM images showing the morphology of (**a**) NF_0.6_, (**b**) NF_1.8_, (**c**) NF_3.6_, (**d**) NF_6.0_, (**e**) NF_7.2_, and (**f**) cross-section SEM image of NF_7.2_.

**Figure 4 polymers-15-03965-f004:**
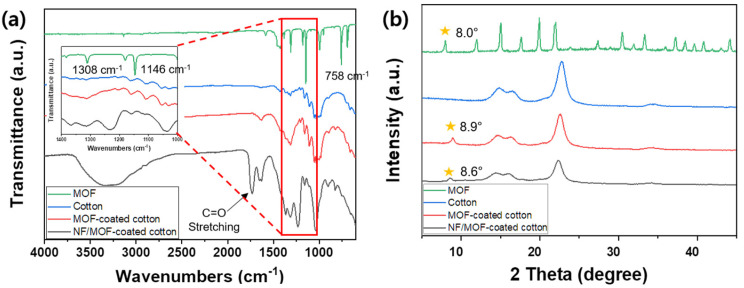
(**a**) FT-IR spectra and (**b**) XRD profiles of the ZIF-8 (green), cotton (blue), MOF-coated cotton (red), and NF/MOF-coated cotton (black) samples.

**Figure 5 polymers-15-03965-f005:**
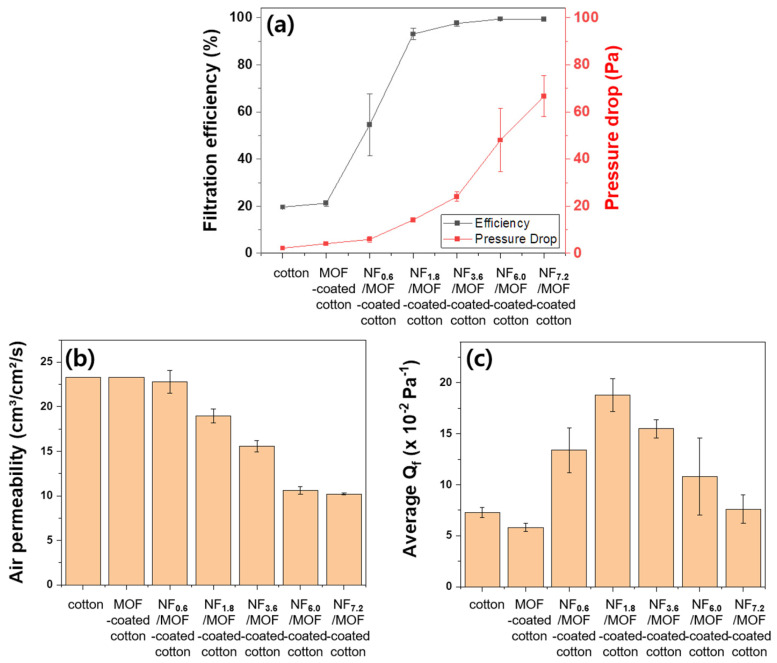
(**a**) Filtration efficiency (left y-axis, black) and pressure drop (right y-axis, red), (**b**) air permeability, and (**c**) average Q_f_ of cotton, MOF-coated cotton, and CA NF/MOF-coated cotton fabricated with different amounts of electrospun CA solutions.

**Figure 6 polymers-15-03965-f006:**
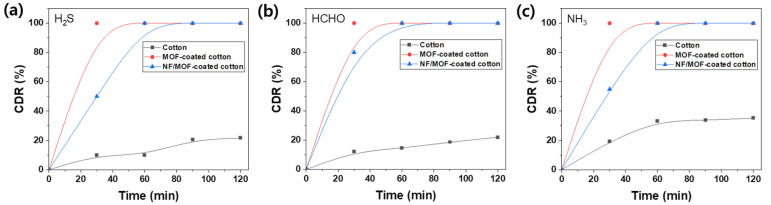
CDR as a function of exposure time for (**a**) H_2_S, (**b**) formaldehyde, and (**c**) NH_3_ measured with pristine cotton fabric (black), MOF-coated cotton fabric (red), and CA NF_1.8_/MOF-coated cotton fabric membranes (blue).

**Figure 7 polymers-15-03965-f007:**
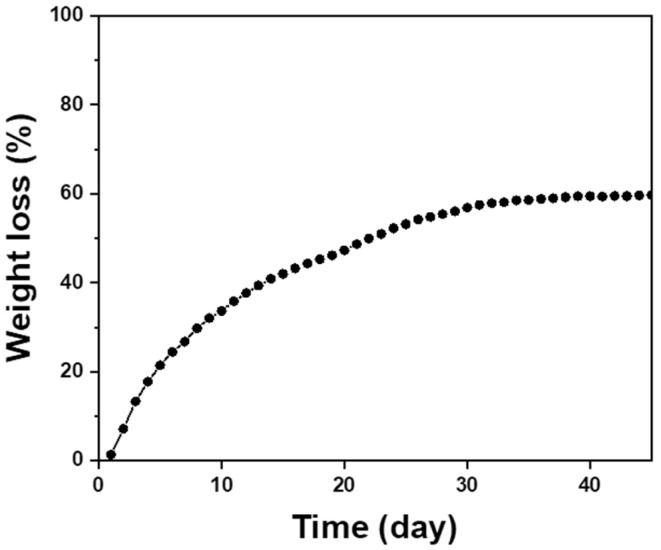
Biodegradability of the NF_1.8_/MOF-coated cotton fabric membrane as a function of time.

## Data Availability

The datasets generated during and/or analyzed during the current study are available from the corresponding author on reasonable request.

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
