# Peer review of "Biodegradable Nanofiber/Metal–Organic Framework/Cotton Air Filtration Membranes Enabling Simultaneous Removal of Toxic Gases and Particulate Matter"

_polymers, 2023, doi:10.3390/polym15193965_

Round 1

Reviewer 1 Report

1.      Keywords: choice one single word keywords such as filtration, adsorption and so on.

2.      Introduction: the authors should improve it in order to highlight the novelty of the work. Moreover, are there similar systems already designed? Which are the advantages of the proposed system?

3.      Characterizations: re-organize this section by dividing it in some short paragraphs on the basis of the kind of analysis (such as microscopy, spectroscopy, permeability and so on). The choice of reporting all of them in one section make quite hard to read it.

4.      Is there a specific correlation between the [ligand] to [Zn2+] ratio and the shape and dimensions of MOF particles? This part should be better discussed.

5.      Is it crucial to control the fibre diameter for the proposed CA based systems? It seems that there are no specific details concerning the dimensions of fibers

6.      The quality of figure 4 should be improved. They are quite blurry and it results hard to clearly read the data reported Moreover, are the data statistically significant? Are there significant differences between the samples?

7.      How can the shape, dimension and the presence itself of MOF affect the gas removal capacity of the designed systems? Are there physical or chemical interactions?

8.      Figure 6: what about the degradation of cotton, CA and cotton MOF? Which one mostly contributed to the degradation?

Minor editing of English language required

Author Response

Response to comments

Reviewer 1

[Response] We thank the reviewer for his or her careful and thoughtful review, and positive and constructive comments to improve the manuscript. We carefully revised the manuscript to fully address the reviewer’s comments.

  1. Keywords: choice one single word keywords such as filtration, adsorption and so on.

Response   We have revised the keywords as following: Nanofiber; Cellulose; Biodegradable; Metal Organic Frameworks; Filtration (Page 1, Line 33)

  1. Introduction: the authors should improve it in order to highlight the novelty of the work. Moreover, are there similar systems already designed? Which are the advantages of the proposed system?

Response Similar membrane has been reported recently: the multifunctional fibrous membrane for removing particulate matters and toxic gases was demonstrated by employing simultaneous electrospinning/electrospray of poly(vinyl alcohol) and MOF onto poly(ethylene terephthalate) nonwoven fabric. (Macromol. Rapid Commun. 2022, 43, 2100648). However, it does not only require complex co-electrospinning system, but also in the process, the adsorbed MOF particles can be desorbed in a variety of conditions. More importantly, the system is not biodegradable, which is not preferred as a next generation material platform.

In the current submission, we report the fabrication route of a biodegradable cellulose-based filtration membrane via the direct growth of ZIF-8 from the surface of cotton fabric and subsequent electrospinning of cellulose acetate (CA) solution to form a CA NF/MOF-coated cotton fabric membrane that can filter particulate matter and toxic gases simultaneously, which should be significant improvement. We highlighted the novelty of our work in Introduction with citing the reference:

“The multifunctional fibrous membrane for removing particulate matters and toxic gases was demonstrated by employing simultaneous electrospinning/electrospray of poly(vinyl alcohol) and MOF onto poly(ethylene terephthalate) nonwoven fabric [4]. However, it does not only require complex co-electrospinning system, but also in the process, the adsorbed MOF particles can be desorbed in a variety of conditions. More importantly, the system is yet biodegradable, which is not preferred as a next generation material platform. Herein, we report the fabrication route of a biodegradable cellulose-based filtration membrane via the direct growth of ZIF-8 from the surface of cotton fabric and subsequent electrospinning of cellulose acetate (CA) solution to form a CA NF/MOF-coated cotton fabric membrane that can filter particulate matter and toxic gases simultaneously (Figure 1).” (Page 2, Line 86)

Added Reference

  • Kim et al. Rational Process Design for Facile Fabrication of Dual Functional Hybrid Membrane of MOF and Electrospun Nanofiber towards High Removal Efficiency of PM5 and Toxic Gases. Macromol. Rapid Commun. 2022, 43, 2100648.

  1. Characterizations: re-organize this section by dividing it in some short paragraphs on the basis of the kind of analysis (such as microscopy, spectroscopy, permeability and so on). The choice of reporting all of them in one section make quite hard to read it.

Response

Thank you for your suggestion. We have divided characterization section to six sections as the reviewer suggested;

2.2. Microscopic characterization

2.3. Spectroscopic characterization

2.4. Assessment of air permeability

2.5. Evaluation of gas adsorption capability

2.6. Assessment of biodegradability.

  1. Is there a specific correlation between the [ligand] to [Zn2+] ratio and the shape and dimensions of MOF particles? This part should be better discussed.

Response

When the ratio of [ligand] to [Zn2+] is adjusted, it is possible to attain ZIF-8 particles with a controlled shape. In this case, the metal ion interacts with the ligand effectively to form the square-shaped particle structure of ZIF-8. If the amount of the ligand is larger than that of the metal ion, we found that irregular particle shape was obtained. The shape of these irregular particles was varied depending on the ratio of ligand to metal during the synthesis process. For example, Ramos et al. reported that the elliptical shape was observed at the ratio of 8, which agrees well with our results. (Pharmaceutics 2022, 14, 2546.) We would like to emphasize that the main purpose of our work is attaining the capability for the gas removal. Therefore, the particles with small size are preferred for maximizing the surface area. At the ratio of 2, it is apparent that the spherical ZIF-8 particles with 1-2 μm, which further showed outstanding gas adsorption capability.

To reflect the reviewer’s comment, we have now added the sentences in the main text with citing the references mentioned above:

“The morphology was changed to a disc-like or elliptical structure as the ratio was increased [26]. In addition, at the ratio higher than 4, the size of disc-like particles gradually decreases, which is in good agreement with the literature [27]. For the applicability to remove gas species, [ligand]/[Zn2+] was set to 2.0 to attain a small spherical MOF structure and maximize the adsorption capability and the surface area.” (Page 5, Line 207)

Added references:

  • Ramos et al., ZIF-8 and Its Magnetic Functionalization as Vehicle for the Transport and Release of Ciprofloxacin. Pharmaceutics 2022, 14, 2546.
  • Watanabe et al., Synthesis of zeolitic imidazolate framework-8 particles of controlled sizes, shapes, and gate adsorption characteristics using a central collision-type microreactor. Eng. J. 2017, 313, 724.

  1. Is it crucial to control the fibre diameter for the proposed CA based systems? It seems that there are no specific details concerning the dimensions of fibers

Response

Generally, the fiber diameter affects the porosity and the specific surface area. The fiber diameter can be easily controlled through varying the concentration of an electrospun polymer solution. The concentration affects the diameter of the fiber has been demonstrated by many literatures. (Sci. Rep. 2018, 8, 15725; Polymers 2020, 12, 2520) As reviewer commented, the fiber diameter will affect the filtration efficiency; a smaller fiber diameter leads to a smaller pore size, eventually increasing the filtration efficiency. However, it should be noted that the fiber diameter affects the filtration system but is less critical compared to the thickness of the nanofiber mat. The thickness of nanofiber mat is the most significant parameter for filtration system. (Build. Environ. 2020, 170, 106628.; Macromol. Rapid Commun. 2022, 43, 2100648.) Therefore, in the current filtration system, NF/MOF/cotton, we have focused on the effect of the volume of electrospun solution, which affects thickness of mat. To reflect the reviewer’s comment, we have added the following paragraph with appropriate reference:

“On the MOF/cotton membrane, a CA solution was electrospun at fixed concentration with different amounts of solution from 0.6 mL to 7.2 mL. Although the concentration affect the fiber diameter in the electrospinning process,[28, 29] in terms of the filtration efficiency, the thickness of the nanofiber mat is more crucial.[4,30] Thus, we focused on the effect of the thickness of the nanofiber mat fabricated at the fixed concentration of the polymer solution.” (Page 6, Line 224)

Added Reference

  • Lee et al. Readily Functionalizable and Stabilizable Polymeric Particles with Controlled Size and Morphology by Electrospray. Rep. 2018, 8, 15725.
  • Lee et al. Electrospinning/electrospray of ferrocene containing copolymers to fabricate ROS-responsive particles and fibers. Polymers 2020, 12, 2520.
  • Bian et al. Influence of fiber diameter, filter thickness, and packing density on PM2. 5 removal efficiency of electrospun nanofiber air filters for indoor applications. Environ. 2020, 170, 106628.
  • Kim et al. Rational Process Design for Facile Fabrication of Dual Functional Hybrid Membrane of MOF and Electrospun Nanofiber towards High Removal Efficiency of PM5 and Toxic Gases. Macromol. Rapid Commun. 2022, 43, 2100648.

  1. The quality of figure 4 should be improved. They are quite blurry and it results hard to clearly read the data reported Moreover, are the data statistically significant? Are there significant differences between the samples?

Response

Thank you for your constructive comments. To reflect the reviewer’s comment, we updated the figure with the version having higher resolution. Regarding the data statistics, All the reported data was obtained by averaging five experiment results, and standard deviation value was provided with the error bars. We note that the individual experiment results were not significantly different from each other. The margin of error is acceptable, and the resulting trends are reasonable.

Figure 5. (a) Filtration efficiency (left y-axis) and pressure drop (right y-axis), (b) air permeability, and (c) average Qf of cotton, MOF-coated cotton, and CA NF/MOF-coated cotton fabricated with different amounts of electrospun CA solutions.

             To reflect the reviewer’s comment, we added the sentence in the characterization section:

             “Five measurements were performed, and the average and the standard deviation values were reported.” (Page 4, Line 135)

  1. How can the shape, dimension and the presence itself of MOF affect the gas removal capacity of the designed systems? Are there physical or chemical interactions?

Response

MOF is known for its versatile design capabilities depending on the target gases to be adsorbed. Due to MOF's inherent porosity, it possesses high gas removal properties by gas adsorption. The structure and pore size of MOFs play a significant role in their capacity for gas adsorption. Both physical and chemical interactions are involved in this context. There are three important factors of MOFs for the gas removal property. First, the shape and size determine the surface area, which is directly related to the adsorption capability. Second, the pore size should match the size of target gas molecules, resulting in the selectivity. Third, the type of metal ion and ligand, i.e. chemical structure, can affect the capacity and selectivity. With these parameters, both physical and chemical interactions are essential for MOFs to adsorb gases effectively as all the possible interactions, e.g. van der Waals forces, dispersion forces, and the electron donation from gas molecule to the metal species in MOF are feasible to interact with the gas molecules.

The advantages of the MOF for gas adsorption are quite well known. We have emphasized these aspects in Introduction:

“The metal organic framework (MOF) is a widely used material as an adsorbent for different gas molecules owing to its high specific surface area and porosity [20]. Among the various MOF materials, zeolitic imidazolate framework (ZIF) has a high potential for the removal or storage of specific gas molecules [21,22].” (Page 2, Line 77)

  1. Figure 6: what about the degradation of cotton, CA and cotton MOF? Which one mostly contributed to the degradation?

Response

In the main text, it was described that the weight ratio of the components in CA/MOFs/Cotton is 16.2 : 2.8 : 81.0 for NF, MOF, and cotton, respectively. It indicates that the biodegradability is mainly originated from cotton fabric as it is the majority in the system. Considering that the main component of cotton is cellulose, in our material system, the cellulose derivatives make up approximately 97% of the filtration membrane. Therefore, it is that cellulose-based materials mostly contribute to the degradation.

We thank the reviewer 1 for the valuable and constructive comments.

Reviewer 2 Report

The objective of this research is to develop a method for producing biodegradable filtration membranes that can effectively remove both particulate matter and toxic gases. The membrane is composed of cellulose-based nanofibers for particulate matter filtration and metal organic framework-coated cotton fabric for toxic gas removal. It has a high filtration efficiency, air permeability, and pressure drop. Additionally, it exhibits exceptional gas adsorption efficiencies for harmful gases. The membrane is highly biodegradable and can be used in personal protective equipment.

The research outline is clear, and the results cover proper data. However, during the review process, a major comment was raised regarding the layout of the results and discussion section. As the core part of the manuscript, this section needs to be organized clearly with proper subtitles. It is recommended that the authors discuss the results section first and then discuss the data, value, and findings and how they impact the field. The authors should reorganize the layout of this section, bring the proper figures to the right positions, focus on data analysis first, and then discussion, and rewrite the contents for this section.

In summary, the authors need to reorganize how they present their data analysis part and discussion, add some support figures from their supplementary section, and clarify some missing details in their whole manuscript.

Besides this major concern, the reviewer listed other questions that arose from reading the paper. The authors are encouraged to address these questions in the revised work. Good luck!

·       Line 18 & 19: Prior to introducing abbreviations, it is recommended to list the full names. eg: PM and ZIF-8 here, NF in line 23

·       Line 64: Can electrospinning be applied on the cotton fiber nanofabrication?

·       In Figure 1, it is suggested to use a similar pattern for MOF layers in both figures and remove the dots that indicate Cotton Fabric and CA nanofibers or use arrows to avoid confusion.

·       - To support the selection of 125 Pa, it is suggested to cite a standard or reference in line 117.

·       - Section 2.3 raises the question of the difference between padding and drying and why two different conditions were chosen to dry/pad immersed cotton fabrics.

·       - The supplementary figures (Figure S1 and Figure S2) are requested, and it is suggested to bring Figure S2 back to the main manuscript since it is related to the morphology of MOF/cotton fabric.

·       - It is recommended to provide each subsection in Section 3 with an appropriate subtitles to divide the results into several sections. It is not appropriate to list section 3.1 with the name "subsection."

·       - The discussion part in lines 185-197, including why the current setup was chosen, should be moved back after the result analysis, as the discussion part

·         Line 210-211: I don’t worry about the gap since the fabric cotton should provide enough spots for the electrospun fiber’s deposition. On the other hand, I am concerned about the adhesion between those two layers, do you have the data or any designed experiments to display the adhesion of those two layers?

·       Line 220 & 222: What do you want to say here? Inthe current version, it is written as : “although these peaks were barely observedin MOF/cotton….1310 peak was observed clearly in MOF/cotton..”?? Clarification is needed here

·       - It is suggested to insert a zoom-in version of the curves in the region of characterization peak (i.e., the region between 1310-1220) and mark the peak you mentioned above in the inserted figure.

·       - The discussion part in lines 243-253 should be moved back after the result analysis.

·         Line 269: What did you learn, or cite from reference 31, to help your team determine that NF1.8/MOF-coated component is the optimal ratio?

·       - In lines 276-280, it is recommended to move the discussion back and provide justification or other references for the mechanism by which MOF/cotton can remove three different types of toxic gases.

·       - In Line 286: What’s the reason for the lower CDR in the NF/MOF group compared to the MOF group among all three model gases in the first 30 minutes?

·       - The discussion part in lines 298-300 should be moved back.

Author Response

Reviewer 2

The objective of this research is to develop a method for producing biodegradable filtration membranes that can effectively remove both particulate matter and toxic gases. The membrane is composed of cellulose-based nanofibers for particulate matter filtration and metal organic framework-coated cotton fabric for toxic gas removal. It has a high filtration efficiency, air permeability, and pressure drop. Additionally, it exhibits exceptional gas adsorption efficiencies for harmful gases. The membrane is highly biodegradable and can be used in personal protective equipment. The research outline is clear, and the results cover proper data.

However, during the review process, a major comment was raised regarding the layout of the results and discussion section. As the core part of the manuscript, this section needs to be organized clearly with proper subtitles. It is recommended that the authors discuss the results section first and then discuss the data, value, and findings and how they impact the field. The authors should reorganize the layout of this section, bring the proper figures to the right positions, focus on data analysis first, and then discussion, and rewrite the contents for this section. In summary, the authors need to reorganize how they present their data analysis part and discussion, add some support figures from their supplementary section, and clarify some missing details in their whole manuscript.

Besides this major concern, the reviewer listed other questions that arose from reading the paper. The authors are encouraged to address these questions in the revised work. Good luck!

[Response] We thank the reviewer for his or her careful and thoughtful review, all insightful and constructive comments to improve the manuscript. We do not disagree to the reviewer’s point that the main text needs to be reorganized. To address the reviewer’s comment, we have divided the Materials and Methods section to 8 sub-sections, and divided the Results and Discussion section to 6 sections. In the Results and Discussion section, we first described the results, then added discussions, as the reviewer suggested. Also, we brought one of the supporting figures to the main text. Lastly, we have modified several sentences to make the flow better. With the current submission, we would like to emphasize that our results should be impactful in the field of nanofibers and application for air remediation. Undoubtedly, we have largely modified the manuscript to fully address the reviewer’s comments.

  1. Line 18 & 19: Prior to introducing abbreviations, it is recommended to list the full names. eg: PM and ZIF-8 here, NF in line 23

Response

Thank for the comment. Current manuscript provides the full names and abbreviations: particulate matter (PM), zeolitic imidazolate framework (ZIF-8), nanofiber (NF).

  1. Line 64: Can electrospinning be applied on the cotton fiber nanofabrication?

Response

The sentence pointed out by the reviewer, in fact, means that the cotton fabric itself cannot be utilized as a filtration media without the help of well-defined nanofiber mat. Therefore, we utilized the nanofiber mat to fabricate a filtration media with cotton fabric, which is used as a support as well as a template for MOF deposition. Our design was successful for the multifunctional membrane that can simultaneously remove particulate matters and toxic gases.

  1. In Figure 1, it is suggested to use a similar pattern for MOF layers in both figures and remove the dots that indicate Cotton Fabric and CA nanofibers or use arrows to avoid confusion.

Response

Thank you for this constructive comment. We have revised the Figure 1 as reviewer suggested as followed:

Figure 1. Schematic diagram displaying the fabrication of the multifunctional hybrid CA NF/MOF-coated cotton membrane for the simultaneous filtration of toxic gas molecules and particulate matter.

  1. To support the selection of 125 Pa, it is suggested to cite a standard or reference in line 117.

             Response

Thank you for this constructive comment. We have added the proper reference to provide the standard regulation for air permeability characterization:

“Air permeability was estimated using FX 3300 Air Permeability Tester â…¢ (TEXTEST, Switzerland) by measuring an air flow rate perpendicularly passing through the sample (effective area = 38 cm2) under an air pressure difference of 125 Pa.[23]” (Page 4, Line 130)

Added reference

- ASTM, D. 737-18. Standard Test Method for Air Permeability Of Textile Fabrics 2023.

  1. Section 2.3 raises the question of the difference between padding and drying and why two different conditions were chosen to dry/pad immersed cotton fabrics.

             Response

The immersion times for the ligand and zinc solutions were the results of our optimization process for the reaction parameters. Immersing the fabric in the ligand solution for 3 hours allowed the effective interaction of ligand molecules with the surface of the cotton fabric, ensuring the uniform distribution of the ligand and providing an opportunity to achieve all necessary parts for MOF formation. Immersion in the zinc solution for 5 hours also fully allow the interaction between zinc and the ligand, leading to the effective formation of the MOF structure. This extended immersion period allowed for zinc ion to react sufficiently on the fabric surface over time, leading to the completion of MOF formation and improving its crystallinity. Both immersion times were selected to maintain the reactions until they reach equilibrium.

Drying time (24 hrs) was chosen to secure the time required for MOF particles to be formed and stabilized completely. This extended drying period promoted the full formation of the MOF structure, increasing the stability. During the drying period, moisture was also effectively removed, which is crucial as moisture during MOF synthesis can induce unwanted chemical reactions or structural deformations in the MOF. This, in turn, improves MOF performance and stability, while also aiding in the reproducibility of the experiments.

  1. The supplementary figures (Figure S1 and Figure S2) are requested, and it is suggested to bring Figure S2 back to the main manuscript since it is related to the morphology of MOF/cotton fabric.

             Response

Thank you for this constructive comment. We have attached supplementary files, and we have moved Figure S2 to the main manuscript as Figure 2.

  1. It is recommended to provide each subsection in Section 3 with an appropriate subtitles to divide the results into several sections. It is not appropriate to list section 3.1 with the name "subsection."

             Response

Thank you for your suggestion. We have divided characterization section to six sections as the reviewer suggested;

2.2. Microscopic characterization

2.3. Spectroscopic characterization

2.4. Assessment of air permeability

2.5. Evaluation of gas adsorption capability

2.6. Assessment of biodegradability.

  1. The discussion part in lines 185-197, including why the current setup was chosen, should be moved back after the result analysis, as the discussion part.

Response

Thank you for this constructive comment. We moved the paragraph back after the morphology analysis as reviewer suggested. In addition, we modified the paragraph and added more sentence to make the flow more smooth:

“Commercial filters made from synthetic polymers, such as polyethylene and poly-propylene, can be used for filtration applications after treatment with corona charging, one of the most exploited techniques because of its convenience. Under humid or high temperature conditions, however, the charges injected into the dielectric materials can be dissipated easily, resulting in a degradation of the durability of the resulting membrane [31]. Cotton can effectively interact with water molecules. Therefore, despite exhibiting excellent biodegradability, it cannot be used widely as a filter medium because of its high water content. In this perspective, electrospun CA NFs were beneficial as they can impart the filtration capability and circumvent this issue. Nano-sized fibers should compensate for the disadvantages of cotton fabric by improving the filtration efficiency much higher than cotton fabrics because of its extremely small pores in the NF membrane. It should be noted that the high filtration efficiency also can result in low air permeability and high-pressure drop, essential factors in filtration performance. Therefore, optimizing the filtration efficiency and air permeability is essential, which is feasible with the control of the thickness of the nanofiber mat, which is one of the most significant parameters to affect the performance of air filtration membranes.” (Page 6, Line 242)

  1. Line 210-211: I don’t worry about the gap since the fabric cotton should provide enough spots for the electrospun fiber’s deposition. On the other hand, I am concerned about the adhesion between those two layers, do you have the data or any designed experiments to display the adhesion of those two layers?

Response

We fully understand the reviewer’s concern about adhesion between two layers. However, considering the use of NF/cotton filter, the adhesion is not the critical issue for this filtration membrane. The image below is a typical layer structure of a facemask. The membrane filter we show can be applied as a middle layer, rather than the outer layer. Therefore, even if the detachment of NFs layer occurred in NF/cotton filter, the outer layer will hold the detachment.

Figure R1. The layer structure of the facemask.

In addition, the nanofiber layer is expected to be attached to cotton fabric due to its small size and the interaction with the MOF-coated cotton. The polar cellulose-based nanofiber can interact with polar MOF-coated cotton surface, and its very small size can maximize the contact area to the surface. As shown in Figure R2 below, the membrane was not separated unless force is applied. Therefore, the adhesion issue between NFs and cotton is not significant for the PPE application in the current study.

Figure R2. The photograph showing that the membrane is not separated and is stable against the force applied.

  1. Line 220 & 222: What do you want to say here? In the current version, it is written as : “although these peaks were barely observed in MOF/cotton….1310 peak was observed clearly in MOF/cotton..”?? Clarification is needed here

Response

In the FT-IR, the peak at 1310 cm-1 which is assigned MOFs peak was barely observed in the MOFs/cotton, and NF/MOFs/cotton peak, which is because the small amount of MOFs was incorporated in MOFs-coated cotton fabric relative to the amount of cotton and CA NFs. This is the reason why we have further provided the XRD patterns. We revised the sentence to clarify the meaning in current revised manuscript.

“These characteristic peaks were barely observed in the spectra of MOF/cotton and NF/MOF/cotton membranes, which are expected as the relative amount of MOF in the composite system is very small compared to cotton and NF: compared to cotton fabric, approximately 2.8 wt% of MOF particles were formed, resulting in a very low intensity of MOF in the FT-IR spectra.” (Page 7, Line 267)

  1. It is suggested to insert a zoom-in version of the curves in the region of characterization peak (i.e., the region between 1310-1220) and mark the peak you mentioned above in the inserted figure.

Response

Thank you for this constructive comment. We have added the zoom-in version of the characterization in Figure 4a. As mentioned in the manuscript, the peaks around 1308 cm-1 and 1146 cm-1 are the peaks due to the imidazole group.

Figure 4. (a) FT-IR spectra and (b) XRD profiles of the ZIF-8 (green), cotton (light blue), MOF-coated cotton (red), and NF/MOF-coated cotton (black) samples

  1. The discussion part in lines 243-253 should be moved back after the result analysis.

Response

Thank you for this constructive comment. We moved the sentences to the end of the subsection, and modified some sentences for better flow:

“Recently, a dual-functional hybrid filtration system consisting of MOF and NF was reported for the removal of particulate matter and toxic gases in the air [4]. The study highlighted the importance of finding the most desirable NF, MOF, and non-woven fabric configurations for optimal filtration performance for the practical PPE application. The membrane in the current study also shows comparable air filtration performance, ensuring the effectiveness of the proposed fabrication approach. Therefore, the NF1.8/MOF-coated cotton membrane exhibiting the highest Qf was further assessed for its ability to remove toxic gases.” (Page 8, Line 315)

  1. Line 269: What did you learn, or cite from reference 31, to help your team determine that NF1.8/MOF-coated component is the optimal ratio?

Response

The reference 31 is the report describing the filtration performance of commonly used mask, and it shows an acceptable range of filtration efficiency and air permeability from various mask filters. From this reference, we claim that that our NF1.8/MOFs-coated cotton membrane also exhibits suitable quality factor for effective filtration performance. We modified the sentence to address the reviewer’s comment:

“Therefore, the NF1.8/MOF-coated cotton membrane exhibiting the highest Qf was further assessed for its ability to remove toxic gases.” (Page 8, Line 320)

  1. In lines 276-280, it is recommended to move the discussion back and provide justification or other references for the mechanism by which MOF/cotton can remove three different types of toxic gases.

Response

Thank you for this constructive comment. We have relocated the two sentences right after the sentence describing the results. Also, for better flow, we modified the sentences:

“MOF is highly effective in capturing these gases because of its active sites and large surface area [38]. In particular, ZIF-8, which was grown as an MOF on the cotton fabric, is used widely for adsorbing harmful gas species [39]. As expected, MOF-coated cotton and NF1.8/MOF-coated cotton membranes exhibited a CDR of approximately 99% for all three gases after 60 min.” (Page 9, Line 335)

  1. In Line 286: What’s the reason for the lower CDR in the NF/MOF group compared to the MOF group among all three model gases in the first 30 minutes?

Response

The results show that the adsorption behavior is mainly dependent on the position of MOF layer. The MOF of the MOF-coated cotton is directly exposed to the gas molecules, resulting in effective and immediate adsorption. However, for the gas molecules to reach MOF in NF/MOF-coated cotton membrane, the gas molecules should penetrate the nanofiber mat layer. During the penetration, the gas molecules can interact with CA NFs, requiring a certain period of time for the adsorption to the MOF. Therefore, the faster adsorption behavior of MOF-coated cotton than NF/MOF-coated cotton should be acceptable.

  1. The discussion part in lines 298-300 should be moved back.

Response

Thank you for this constructive comment. We moved the sentence after the results description, and also modified some sentences for better flow:

“The majority of the NF1.8/MOF-coated cotton membrane is a cellulose-based material because the calculated weight ratio of the components is 16.2 : 2.8 : 81.0 (NF, MOF, and cotton), suggesting that the hybrid membrane exhibits adequate biodegradability.” (Page 10, Line 348)

“The cotton-based membrane filter is expected to exhibit biodegradability because cotton is readily degraded by microorganisms in the soil when landfilled. Venditti et al. reported that approximately 40% of cotton fabric was degraded after 40 days [41]. Li et al. reported that the degradation efficiency of the cotton fabric varied in the range of 50–77% after 90 days of treatment [42]. Our results agree well with the literature, therefore, the proposed route effectively fabricates a cellulose-based biodegradable dual-functional membrane to remove particulate matter and toxic gases simultaneously.” (Page 9, Line 354)

We thank the reviewer 2 for the valuable and constructive comments.

Reviewer 3 Report

Dear Authors

The authors in this study developed a biodegradable dual-functional hybrid membrane through the route, combining the direct growth of MOF on cotton fabric and subsequent electrospinning of a CA solution. The ZIF-8 was grown from the surface of cotton fabric by treatment with an organic precursor solution and subsequent immersion in the metal ion solution. The CA NF was electrospun onto the MOF-coated cotton fabric with different amounts of CA solution to control the membrane porosity. The thickness of the NF layer largely affects the filtration performance for particulate matter, as explored with the assessment of filtration efficiency, pressure drop, and air permeability. With a thicker NF layer, the filtration efficiency and the pressure drop tended to increase, even though the air permeability decreased. As all of these parameters are in a trade-off relationship, Qf was assessed to find the optimal thickness of the NF layer: NF1.8/MOF-coated fabric exhibited the highest Qf value (18.8 × 10−2 Pa−1) with an air permeability of 19.0  cm3/cm2/s, pressure drop of 14.2 Pa, and filtration efficiency of 93.1%. The MOF-coated cotton and NF1.8/MOF-coated cotton membrane showed CDR values higher than 99% for H2S, formaldehyde, and NH3, confirming the achievement of the dual functions of toxic gas removal and the filtration of particulate matter in the membrane system. Finally, the NF1.8/MOF-coated cotton fabric membrane showed reasonable biodegradability with a weight loss of 62.5% after 45 days, which is acceptable to consider the system as a biodegradable material. The presented strategy combining the growth of MOF and electrospinning cellulose-based NF should offer an eco-friendly approach to achieve lightweight PPE filters that simultaneously remove particulate matter and toxic gases sustainably.

The approach and research design are quite simple and applicable. Maybe a few comments below are recommended to consider during the revision process before considering manuscript publication. 

General comment

The durability of the developed filtration membranes should be considered to show the benefits of the developed approach in this study.

Specific comments

Abstract: The authors should mention the full name of any expression the first time it appeared in the text, such as PM filtration, ZIF-8, and NF layer. 

2.2. Characterization

The authors should use subtitles to make this section clearer and easy to follow. Equations used should be numbered. 

2.4. Electrospinning of CA NFs on cotton fabric

The authors must clearly state how to mount the MOF-cotton fabrics on the cylinder-type metallic drum collector.

A major revision is recommended. 

A minor revision is recommended. 

Author Response

Reviewer 3

The approach and research design are quite simple and applicable. Maybe a few comments below are recommended to consider during the revision process before considering manuscript publication. 

[Response] We thank the reviewer for his or her careful and thoughtful review, and positive and constructive comments to improve the manuscript. We carefully revised the manuscript to fully address the reviewer’s comments.

  1. 1.    The durability of the developed filtration membranes should be considered to show the benefits of the developed approach in this study.

Response

Thank for your constructive comment. We fully understand the concerns regarding the durability of the developed filters and understand that this is an important parameter for face mask applications. However, the suggested membrane has the significant property and biodegradability. Even if the filter has durability, the facemask is typically disposable, i.e. easily discarded after use. We have focused on the enhancement of the biodegradability to reduce the environmental issues induced by the disposed masks. The cellulose based biodegradable filter is a perfect candidate. Moreover, our membrane shows very high filtration performance as well as toxic gases removal. Even though the durability was not deeply considered in the current study, the filtration performance assessed through ISO standard experiments should be invaluable for next generation PPE application.

  1. 2.   Abstract: The authors should mention the full name of any expression the first time it appeared in the text, such as PM filtration, ZIF-8, and NF layer. 

Response

Thank for the comment. Current manuscript provides the full names and abbreviations: particulate matter (PM), zeolitic imidazolate framework (ZIF-8), nanofiber (NF).

  1. 3.    Characterization

The authors should use subtitles to make this section clearer and easy to follow. Equations used should be numbered. 

Response

Thank you for your suggestion. We have divided characterization section to six sections as the reviewer suggested;

2.2. Microscopic characterization

2.3. Spectroscopic characterization

2.4. Assessment of air permeability

2.5. Evaluation of gas adsorption capability

2.6. Assessment of biodegradability.

In addition, all equations were numbered as suggested:

η = (Cup – Cdown)/Cup.                                       (1)

`  Qf = –ln(1 – η)/∆P                                        (2)

CDR (%) = [(Ci – Cf)/Ci] × 100                         (3)

Dt = [((CO2)T – (CO2)B)/(CO2)Th] × 100            (4)

  1. 4.    Electrospinning of CA NFs on cotton fabric

The authors must clearly state how to mount the MOF-cotton fabrics on the cylinder-type metallic drum collector.

Response

Mounting the fabric is general process in electrospinning. Using the scotch tape, substrate was attached on metallic drum collector. To reflect the reviewer’s comment, we have added the following sentence:

“The cotton fabric was mounted on the metallic drum collector using the scotch tape.” (Page 5, Line 186)

We thank the reviewer 3 for the valuable and constructive comments.

Round 2

Reviewer 1 Report

The authors have satisfactorily revised the mansucript. I don't have any further comment.

Reviewer 3 Report

Dear Authors

The revised manuscript has been improved according to the corrections made by the authors based on the reviewer-raised comments.

The revised version can be accepted for publication.

Greetings

A minor revision is recommended.